# ADef: an Iterative Algorithm to Construct Adversarial Deformations

**Rima Alaifari**
Department of Mathematics
ETH Zurich
`rima.alaifari@math.ethz.ch`

**Giovanni S. Alberti**
Department of Mathematics
University of Genoa
`alberti@dima.unige.it`

**Tandri Gauksson**
Department of Mathematics
ETH Zurich
`tandri.gauksson@math.ethz.ch`

## Abstract

While deep neural networks have proven to be a powerful tool for many recognition and classification tasks, their stability properties are still not well understood. In the past, image classifiers have been shown to be vulnerable to so-called adversarial attacks, which are created by additively perturbing the correctly classified image. In this paper, we propose the ADef algorithm to construct a different kind of adversarial attack created by iteratively applying small deformations to the image, found through a gradient descent step. We demonstrate our results on MNIST with convolutional neural networks and on ImageNet with Inception-v3 and ResNet-101.

## 1 Introduction

In a first observation in Szegedy et al. (2013) it was found that deep neural networks exhibit unstable behavior to small perturbations in the input. For the task of image classification this means that two visually indistinguishable images may have very different outputs, resulting in one of them being misclassified even if the other one is correctly classified with high confidence. Since then, a lot of research has been done to investigate this issue through the construction of adversarial examples: given a correctly classified image $x$, we look for an image $y$ which is visually indistinguishable from $x$ but is misclassified by the network. Typically, the image $y$ is constructed as $y = x + r$, where $r$ is an *adversarial perturbation* that is supposed to be small in a suitable sense (normally, with respect to an $\ell^p$ norm). Several algorithms have been developed to construct adversarial perturbations, see Goodfellow et al. (2014); Moosavi Dezfooli et al. (2016); Kurakin et al. (2017b); Madry et al. (2018); Carlini & Wagner (2017b) and the review paper Akhtar & Mian (2018).

Even though such pathological cases are very unlikely to occur in practice, their existence is relevant since malicious attackers may exploit this drawback to fool classifiers or other automatic systems. Further, adversarial perturbations may be constructed in a black-box setting (i.e., without knowing the architecture of the DNN but only its outputs) (Papernot et al., 2017; Moosavi-Dezfooli et al., 2017) and also in the physical world (Kurakin et al., 2017b; Athalye & Sutskever, 2017; Brown et al., 2017; Sharif et al., 2016). This has motivated the investigation of defenses, i.e., how to make the network invulnerable to such attacks, see Kurakin et al. (2017a); Carlini & Wagner (2017a); Madry et al. (2018); Tramèr et al. (2018); Wong & Kolter (2018); Raghunathan et al. (2018); Athalye et al. (2018); Kannan et al. (2018). In most cases, adversarial examples are artificially created and then used to retrain the network, which becomes more stable under these types of perturbations.

Most of the work on the construction of adversarial examples and on the design of defense strategies has been conducted in the context of small perturbations $r$ measured in the $\ell^\infty$ norm. However, this is not necessarily a good measure of image similarity: e.g., for two translated images $x$ and $y$, the norm of $x - y$ is not small in general, even though $x$ and $y$ will look indistinguishable if the translation is small. Several papers have investigated the construction of adversarial perturbations not designed

for norm proximity (Rozsa et al., 2016; Sharif et al., 2016; Brown et al., 2017; Engstrom et al., 2017; Xiao et al., 2018).

In this work, we build up on these ideas and investigate the construction of *adversarial deformations*. In other words, the misclassified image $y$ is not constructed as an additive perturbation $y = x + r$, but as a deformation $y = x \circ (\mathrm{id} + \tau)$, where $\tau$ is a vector field defining the transformation. In this case, the similarity is not measured through a norm of $y - x$, but instead through a norm of $\tau$, which quantifies the deformation between $y$ and $x$.

We develop an efficient algorithm for the construction of adversarial deformations, which we call ADef. It is based on the main ideas of DeepFool (Moosavi Dezfooli et al., 2016), and iteratively constructs the smallest deformation to misclassify the image. We test the procedure on MNIST (LeCun) (with convolutional neural networks) and on ImageNet (Russakovsky et al., 2015) (with Inception-v3 (Szegedy et al., 2016) and ResNet-101 (He et al., 2016)). The results show that ADef can succesfully fool the classifiers in the vast majority of cases (around 99%) by using very small and imperceptible deformations. We also test our adversarial attacks on adversarially trained networks for MNIST. Our implementation of the algorithm can be found at `https://gitlab.math.ethz.ch/tandrig/ADef`.

The results of this work have initially appeared in the master's thesis Gauksson (2017), to which we refer for additional details on the mathematical aspects of this construction. While writing this paper, we have come across Xiao et al. (2018), in which a similar problem is considered and solved with a different algorithm. Whereas in Xiao et al. (2018) the authors use a second order solver to find a deforming vector field, we show how a first order method can be formulated efficiently and justify a smoothing operation, independent of the optimization step. We report, for the first time, success rates for adversarial attacks with deformations on ImageNet. The topic of deformations has also come up in Jaderberg et al. (2015), in which the authors introduce a class of learnable modules that deform inputs in order to increase the performance of existing DNNs, and Fawzi & Frossard (2015), in which the authors introduce a method to measure the invariance of classifiers to geometric transformations.

## 2 ADVERSARIAL DEFORMATIONS

### 2.1 ADVERSARIAL PERTURBATIONS

Let $\mathcal{K}$ be a classifier of images consisting of $P$ pixels into $L \geq 2$ categories, i.e. a function from the space of images $X = \mathbb{R}^{cP}$, where $c = 1$ (for grayscale images) or $c = 3$ (for color images), and into the set of labels $\mathcal{L} = \{1, \ldots, L\}$. Suppose $x \in X$ is an image that is correctly classified by $\mathcal{K}$ and suppose $y \in X$ is another image that is *imperceptible* from $x$ and such that $\mathcal{K}(y) \neq \mathcal{K}(x)$, then $y$ is said to be an *adversarial example*. The meaning of imperceptibility varies, but generally, proximity in $\ell^p$-norm (with $1 \leq p \leq \infty$) is considered to be a sufficient substitute. Thus, an *adversarial perturbation* for an image $x \in X$ is a vector $r \in X$ such that $\mathcal{K}(x + r) \neq \mathcal{K}(x)$ and $\|r\|_p$ is small, where

$$\|r\|_p = \left( \sum_{j=1}^{cP} |r_j|^p \right)^{1/p} \quad \text{if } 1 \leq p < \infty, \text{ and} \quad \|r\|_\infty = \max_{j=1,\ldots,cP} |r_j|. \tag{1}$$

Given such a classifier $\mathcal{K}$ and an image $x$, an adversary may attempt to find an adversarial example $y$ by minimizing $\|x - y\|_p$ subject to $\mathcal{K}(y) \neq \mathcal{K}(x)$, or even subject to $\mathcal{K}(y) = k$ for some *target label* $k \neq \mathcal{K}(x)$. Different methods for finding minimal adversarial perturbations have been proposed, most notably FGSM (Goodfellow et al., 2014) and PGD (Madry et al., 2018) for $\ell^\infty$, and the DeepFool algorithm (Moosavi Dezfooli et al., 2016) for general $\ell^p$-norms.

### 2.2 DEFORMATIONS

Instead of constructing adversarial perturbations, we intend to fool the classifier by small deformations of correctly classified images. Our procedure is in the spirit of the DeepFool algorithm. Before we explain it, let us first clarify what we mean by a deformation of an image. The discussion is at first more intuitive if we model images as functions $\xi : [0, 1]^2 \to \mathbb{R}^c$ (with $c = 1$ or $c = 3$) instead

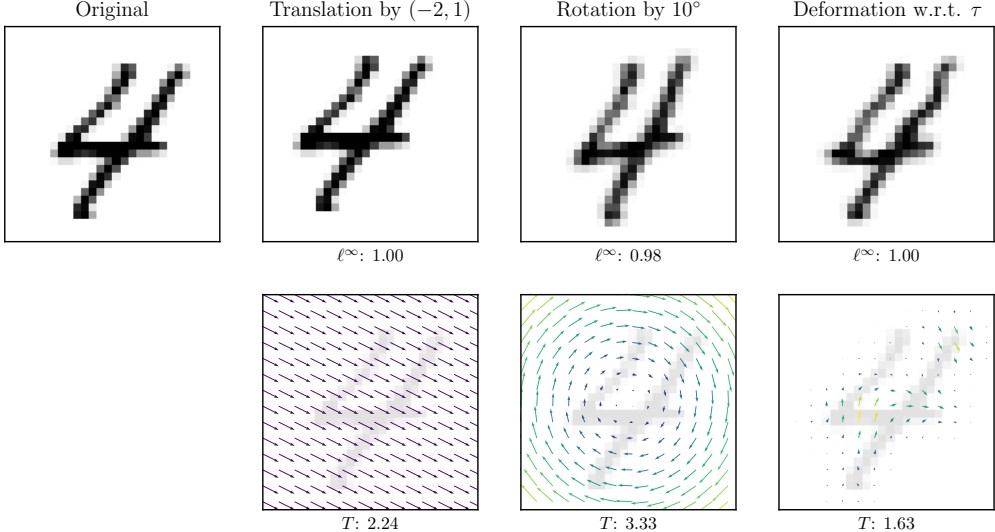

Figure 1: **First row:** The original $28 \times 28$ pixel image from the MNIST database, and the same image translated by $(-2, 1)$, rotated by an angle of $10°$, and deformed w.r.t. an arbitrary smooth vector field $\tau$. The $\ell^\infty$-norm of the corresponding perturbation is shown under each deformed image. The pixel values range from 0 (white) to 1 (black), so the deformed images all lie far from the original image in the $\ell^\infty$-norm. **Second row:** The vector fields corresponding to the above deformations and their $T$-norms (cf. equation (3)).

of discrete vectors $x$ in $\mathbb{R}^{cP}$. In this setting, perturbing an image $\xi : [0, 1]^2 \to \mathbb{R}^c$ corresponds to adding to it another function $\rho : [0, 1]^2 \to \mathbb{R}^c$ with a small $L^p$-norm.

While any transformation of an image $\xi$ can be written as a perturbation $\xi + \rho$, we shall restrict ourselves to a particular class of transformations. A *deformation* with respect to a vector field $\tau : [0, 1]^2 \to \mathbb{R}^2$ is a transformation of the form $\xi \mapsto \xi^\tau$, where for any image $\xi : [0, 1]^2 \to \mathbb{R}^c$, the image $\xi^\tau : [0, 1]^2 \to \mathbb{R}^c$ is defined by

$$\xi^\tau(u) = \xi\left(u + \tau(u)\right) \quad \text{for all } u \in [0, 1]^2,$$

extending $\xi$ by zero outside of $[0, 1]^2$. Deformations capture many natural image transformations. For example, a translation of the image $\xi$ by a vector $v \in \mathbb{R}^2$ is a deformation with respect to the constant vector field $\tau = v$. If $v$ is small, the images $\xi$ and $\xi^v$ may look similar, but the corresponding perturbation $\rho = \xi^v - \xi$ may be arbitrarily large in the aforementioned $L^p$-norms. Figure 1 shows three minor deformations, all of which yield large $L^\infty$-norms.

In the discrete setting, deformations are implemented as follows. We consider square images of $W \times W$ pixels and define the space of images to be $X = \left(\mathbb{R}^{W \times W}\right)^c$. A discrete vector field is a function $\tau : \{1, \ldots, W\}^2 \to \mathbb{R}^2$. In what follows we will only consider the set $T$ of vector fields that do not move points on the grid $\{1, \ldots, W\}^2$ outside of $[1, W]^2$. More precisely,

$$T := \left\{\tau : \{1, \ldots, W\}^2 \to \mathbb{R}^2 \mid \tau(s, t) + (s, t) \in [1, W]^2 \text{ for all } s, t \in \{1, \ldots, W\}\right\}.$$

An image $x \in X$ can be viewed as the collection of values of a function $\xi : [0, 1]^2 \to \mathbb{R}^c$ on a regular grid $\{1/(W+1), \ldots, W/(W+1)\}^2 \subseteq [0, 1]^2$, i.e. $x_{s,t} = \xi(s/(W+1), t/(W+1))$ for $s, t = 1, \ldots, W$. Such a function $\xi$ can be computed by interpolating from $x$. Thus, the deformation of an image $x$ with respect to the discrete vector field $\tau$ can be defined as the discrete deformed image $x^\tau$ in $X$ by

$$x_{s,t}^\tau = \xi\left(\frac{(s, t) + \tau(s, t)}{W + 1}\right), \qquad s, t \in \{1, \ldots, W\}. \tag{2}$$

It is not straightforward to measure the size of a deformation such that it captures the visual difference between the original image $x$ and its deformed counterpart $x^\tau$. We will use the size of the

corresponding vector field, $\tau$, in the norm defined by

$$\|\tau\|_T = \max_{s,t=1,\ldots,W} \|\tau(s,t)\|_2 \tag{3}$$

as a proxy. The $\ell^p$-norms defined in (1), adapted to vector fields, can be used as well. (We remark, however, that none of these norms define a distance between $x$ and $x^\tau$, since two vector fields $\tau, \sigma \in T$ with $\|\tau\|_T \neq \|\sigma\|_T$ may produce the same deformed image $x^\tau = x^\sigma$.)

## 2.3 THE ALGORITHM ADEF

We will now describe our procedure for finding deformations that will lead a classifier to yield an output different from the original label.

Let $F = (F_1, \ldots, F_L) : X \to \mathbb{R}^L$ be the underlying model for the classifier $\mathcal{K}$, such that

$$\mathcal{K}(x) = \arg\max_{k=1,\ldots,L} F_k(x).$$

Let $x \in X$ be the image of interest and fix $\xi : [0,1]^2 \to \mathbb{R}^c$ obtained by interpolation from $x$. Let $l = \mathcal{K}(x)$ denote the true label of $x$, let $k \in \mathcal{L}$ be a target label and set $f = F_k - F_l$. We assume that $x$ does not lie on a decision boundary, so that we have $f(x) < 0$.

We define the function $g : T \to \mathbb{R}, \tau \mapsto f(x^\tau)$ and note that $g(0) = f(x^0) = f(x) < 0$. Our goal is to find a small vector field $\tau \in T$ such that $g(\tau) = f(x^\tau) \geq 0$. We can use a linear approximation of $g$ around the zero vector field as a guide:

$$g(\tau) \approx g(0) + (\mathrm{D}_0 g)\,\tau \tag{4}$$

for small enough $\tau \in T$ and $\mathrm{D}_0 g : T \to \mathbb{R}$ the derivative of $g$ at $\tau = 0$. Hence, if $\tau$ is a vector field such that

$$(\mathrm{D}_0 g)\,\tau = -g(0) \tag{5}$$

and $\|\tau\|_T$ is small, then the classifier $\mathcal{K}$ has approximately equal confidence for the deformed image $x^\tau$ to have either label $l$ or $k$. This is a scalar equation with unknown in $T$, and so has infinitely many solutions. In order to select $\tau$ with small norm, we solve it in the least-squares sense.

In view of (2), we have $\frac{\partial x^\tau}{\partial \tau}|_{\tau=0}(s,t) = \frac{1}{W+1}\nabla\xi\left(\frac{(s,t)}{W+1}\right) \in \mathbb{R}^{c\times 2}$. Thus, by applying the chain rule to $g(\tau) = f(x^\tau)$, we obtain that its derivative at $\tau = 0$ can, with a slight abuse of notation, be identified with the vector field

$$\mathrm{D}_0 g(s,t) = \frac{1}{W+1}\left(\nabla f(x)\right)_{s,t}\nabla\xi\left(\frac{(s,t)}{W+1}\right), \tag{6}$$

where $\left(\nabla f(x)\right)_{s,t} \in \mathbb{R}^{1\times c}$ is the derivative of $f$ in $x$ calculated at $(s,t)$. With this, $(\mathrm{D}_0 g)\,\tau$ stands for $\sum_{s,t=1}^{W}\mathrm{D}_0 g(s,t)\cdot\tau(s,t)$, and the solution to (5) in the least-square sense is given by

$$\tau = -\frac{f(x)}{\sum_{s,t=1}^{W}|\mathrm{D}_0 g(s,t)|^2}\,\mathrm{D}_0 g. \tag{7}$$

Finally, we define the deformed image $x^\tau \in X$ according to (2).

One might like to impose some degree of smoothness on the deforming vector field. In fact, it suffices to search in the range of a smoothing operator $\mathcal{S} : T \to T$. However, this essentially amounts to applying $\mathcal{S}$ to the solution from the larger search space $T$. Let $\alpha = \mathcal{S}(\mathrm{D}_0 g) = \varphi * (\mathrm{D}_0 g)$, where $\mathcal{S}$ denotes the componentwise application of a two-dimensional Gaussian filter $\varphi$ (of any standard deviation). Then the vector field

$$\tilde{\tau} = -\frac{f(x)}{\sum_{s,t=1}^{W}|\alpha(s,t)|^2}\,\mathcal{S}\alpha = -\frac{f(x)}{\sum_{s,t=1}^{W}|\alpha(s,t)|^2}\,\mathcal{S}^2(\mathrm{D}_0 g)$$

also satisfies (5), since $\mathcal{S}$ is self-adjoint. We can hence replace $\tau$ by $\tilde{\tau}$ to obtain a smooth deformation of the image $x$.

We iterate the deformation process until the deformed image is misclassified. More explicitly, let $x^{(0)} = x$ and for $n \geq 1$ let $\tau^{(n)}$ be given by (7) for $x^{(n-1)}$. Then we can define the iteration as

$x^{(n)} = x^{(n-1)} \circ (\mathrm{id} + \tau^{(n)})$. The algorithm terminates and outputs an adversarial example $y = x^{(n)}$ if $\mathcal{K}(x^{(n)}) \neq l$. The iteration also terminates if $x^{(n)}$ lies on a decision boundary of $\mathcal{K}$, in which case we propose to introduce an overshoot factor $1 + \eta$ on the *total* deforming vector field. Provided that the number of iterations is moderate, the total vector field can be well approximated by $\tau^* = \tau^{(1)} + \cdots + \tau^{(n)}$ and the process can be altered to output the deformed image $y = x \circ (\mathrm{id} + (1+\eta)\tau^*)$ instead.

The target label $k$ may be chosen in each iteration to minimize the vector field to obtain a better approximation in the linearization (4). More precisely, for a candidate set of labels $k_1, \ldots, k_m$, we compute the corresponding vectors fields $\tau_1, \ldots, \tau_m$ and select

$$k = \underset{j=1,\ldots,m}{\arg\min} \|\tau_j\|_T.$$

The candidate set consists of the labels corresponding to the indices of the $m$ smallest entries of $F - F_l$, in absolute value.

---

**Algorithm** ADef

---

**Input:** Classification model $F$, image $x$, correct label $l$, candidate labels $k_1, \ldots, k_m$
**Output:** Deformed image $y$

    Initialize $y \leftarrow x$
    **while** $\mathcal{K}(y) = l$ **do**
        **for** $j = 1, \ldots, m$ **do**
            $\alpha_j \leftarrow \mathcal{S}\left(\sum_{i=1}^{c}\left((\nabla F_{k_j})_i - (\nabla F_l)_i\right) \cdot \nabla y_i\right)$
            $\tau_j \leftarrow -\frac{F_{k_j}(y) - F_l(y)}{\|\alpha_j\|_{\ell^2}^2}\mathcal{S}\alpha_j$
        **end for**
        $i \leftarrow \arg\min_{j=1,\ldots,m} \|\tau_j\|_T$
        $y \leftarrow y \circ (\mathrm{id} + \tau_i)$
    **end while**
    **return** $y$

---

By equation (6), provided that $\nabla f$ is moderate, the deforming vector field takes small values wherever $\xi$ has a small derivative. This means that the vector field will be concentrated on the edges in the image $x$ (see e.g. the first row of figure 2). Further, note that the result of a deformation is always a valid image in the sense that it does not violate the pixel value bounds. This is not guaranteed for the perturbations computed with DeepFool.

## 3  EXPERIMENTS

### 3.1  SETUP

We evaluate the performance of ADef by applying the algorithm to classifiers trained on the MNIST (LeCun) and ImageNet (Russakovsky et al., 2015) datasets. Below, we briefly describe the setup of the experiments and in tables 1 and 2 we summarize their results.

**MNIST:** We train two convolutional neural networks based on architectures that appear in Madry et al. (2018) and Tramèr et al. (2018) respectively. The network MNIST-A consists of two convolutional layers of sizes $32 \times 5 \times 5$ and $64 \times 5 \times 5$, each followed by $2 \times 2$ max-pooling and a rectifier activation function, a fully connected layer into dimension 1024 with a rectifier activation function, and a final linear layer with output dimension 10. The network MNIST-B consists of two convolutional layers of sizes $128 \times 3 \times 3$ and $64 \times 3 \times 3$ with a rectifier activation function, a fully connected layer into dimension 128 with a rectifier activation function, and a final linear layer with output dimension 10. During training, the latter convolutional layer and the former fully connected layer of MNIST-B are subject to dropout of drop probabilities $1/4$ and $1/2$. We use ADef to produce adversarial deformations of the images in the test set. The algorithm is configured to pursue any label different from the correct label (all incorrect labels are candidate labels). It performs smoothing by a Gaussian filter of standard deviation $1/2$, uses bilinear interpolation to obtain intermediate pixel intensities, and it overshoots by $\eta = 2/10$ whenever it converges to a decision boundary.

Table 1: The results of applying ADef to the images in the MNIST test set and the ILSVRC2012 validation set. The accuracy of the Inception and ResNet models is defined as the top-1 accuracy on the center-cropped and resized images. The success rate of ADef is shown as a percentage of the correctly classified inputs. The pixel range is scaled to $[0, 1]$, so the perturbation $r = y - x$, where $x$ is the input and $y$ the output of ADef, has values in $[-1, 1]$. The averages in the three last columns are computed over the set of images on which ADef is successful. Recall the definition of the vector field norm in equation (3).

| Model | Accuracy | ADef success | Avg. $\|\tau^*\|_T$ | Avg. $\|r\|_\infty$ | Avg. # iterations |
|---|---|---|---|---|---|
| MNIST-A | 98.99% | 99.85% | 1.1950 | 0.7455 | 7.002 |
| MNIST-B | 98.91% | 99.51% | 1.0841 | 0.7654 | 4.422 |
| Inception-v3 | 77.56% | 98.94% | 0.5984 | 0.2039 | 4.050 |
| ResNet-101 | 76.97% | 99.78% | 0.5561 | 0.1882 | 4.176 |

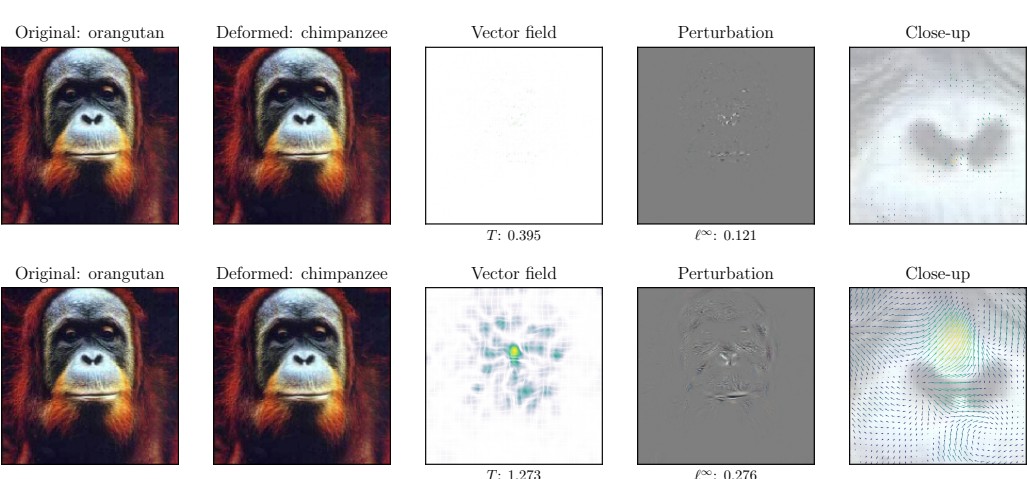

Figure 2: Sample deformations for the Inception-v3 model. The vector fields and perturbations have been amplified for visualization. **First row:** An image from the ILSVRC2012 validation set, the output of ADef with a Gaussian filter of standard deviation 1, the corresponding vector field and perturbation. The rightmost image is a close-up of the vector field around the nose of the ape. **Second row:** A larger deformation of the same image, obtained by using a wider Gaussian filter (standard deviation 6) for smoothing.

**ImageNet:** We apply ADef to pretrained Inception-v3 (Szegedy et al., 2016) and ResNet-101 (He et al., 2016) models to generate adversarial deformations for the images in the ILSVRC2012 validation set. The images are preprocessed by first scaling so that the smaller axis has 299 pixels for the Inception model and 224 pixels for ResNet, and then they are center-cropped to a square image. The algorithm is set to focus only on the label of second highest probability. It employs a Gaussian filter of standard deviation 1, bilinear interpolation, and an overshoot factor $\eta = 1/10$.

We only consider inputs that are correctly classified by the model in question, and, since $\tau^* = \tau^{(1)} + \cdots + \tau^{(n)}$ approximates the total deforming vector field, we declare ADef to be successful if its output is misclassified and $\|\tau^*\|_T \leq \varepsilon$, where we choose $\varepsilon = 3$. Observe that, by (3), a deformation with respect to a vector field $\tau$ does not displace any pixel further away from its original position than $\|\tau\|_T$. Hence, for high resolution images, the choice $\varepsilon = 3$ indeed produces small deformations if the vector fields are smooth. In appendix A, we illustrate how the success rate of ADef depends on the choice of $\varepsilon$.

When searching for an adversarial example, one usually searches for a perturbation with $\ell^\infty$-norm smaller than some small number $\varepsilon > 0$. Common choices of $\varepsilon$ range from $1/10$ to $3/10$ for MNIST

Table 2: Success rates for PGD and ADef attacks on adversarially trained networks.

| Model | Adv. training | Accuracy | PGD success | ADef success |
|---|---|---|---|---|
| MNIST-A | PGD | 98.36% | 5.81% | 6.67% |
| | ADef | 98.95% | 100.00% | 54.16% |
| MNIST-B | PGD | 98.74% | 5.84% | 20.35% |
| | ADef | 98.79% | 100.00% | 45.07% |

Figure 3: Targeted ADef against MNIST-A. **First row:** The original image and deformed images produced by restricting ADef to the target labels $0$ to $8$. The $\ell^\infty$-norms of the corresponding perturbations are shown under the deformed images. **Second row:** The vector fields corresponding to the deformations and their $T$-norms.

classifiers (Goodfellow et al., 2014; Madry et al., 2018; Wong & Kolter, 2018; Tramèr et al., 2018; Kannan et al., 2018) and $2/255$ to $16/255$ for ImageNet classifiers (Goodfellow et al., 2014; Kurakin et al., 2017a; Tramèr et al., 2018; Kannan et al., 2018). Table 1 shows that on average, the perturbations obtained by ADef are quite large compared to those constraints. However, as can be seen in figure 2, the relatively high resolution images of the ImageNet dataset can be deformed into adversarial examples that, while corresponding to large perturbations, are not visibly different from the original images. In appendices B and C, we give more examples of adversarially deformed images.

## 3.2 ADVERSARIAL TRAINING

In addition to training MNIST-A and MNIST-B on the original MNIST data, we train independent copies of the networks using the adversarial training procedure described by Madry et al. (2018). That is, before each step of the training process, the input images are adversarially perturbed using the PGD algorithm. This manner of training provides increased robustness against adversarial perturbations of low $\ell^\infty$-norm. Moreover, we train networks using ADef instead of PGD as an adversary. In table 2 we show the results of attacking these adversarially trained networks, using ADef on the one hand, and PGD on the other. We use the same configuration for ADef as above, and for PGD we use 40 iterations, step size $1/100$ and $3/10$ as the maximum $\ell^\infty$-norm of the perturbation. Interestingly, using these configurations, the networks trained against PGD attacks are more resistant to adversarial deformations than those trained against ADef.

## 3.3 TARGETED ATTACKS

ADef can also be used for *targeted* adversarial attacks, by restricting the deformed image to have a particular target label instead of any label which yields the optimal deformation. Figure 3 demonstrates the effect of choosing different target labels for a given MNIST image, and figure 4 shows the result of targeting the label of lowest probability for an image from the ImageNet dataset.

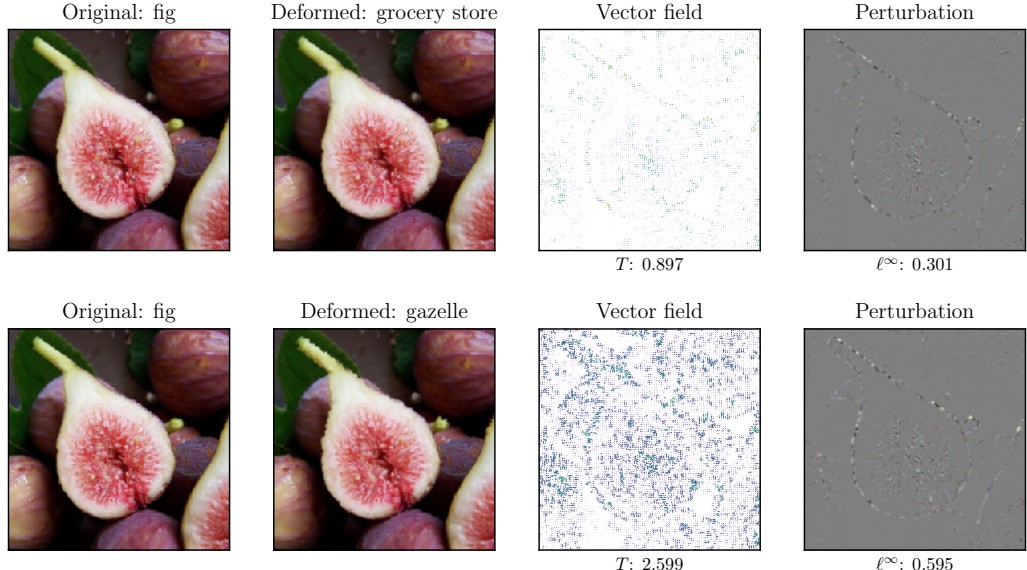

Figure 4: Untargeted vs. targeted attack on the ResNet-101 model. An image from the ILSVRC2012 validation set deformed to the labels of second highest (first row) and lowest (second row) probabilities (out of 1,000) for the original image. The vector fields and perturbations have been amplified for visualization.

# 4 CONCLUSION

In this work, we proposed a new efficient algorithm, ADef, to construct a new type of adversarial attacks for DNN image classifiers. The procedure is iterative and in each iteration takes a gradient descent step to deform the previous iterate in order to push to a decision boundary.

We demonstrated that with almost imperceptible deformations, state-of-the art classifiers can be fooled to misclassify with a high success rate of ADef. This suggests that networks are vulnerable to different types of attacks and that simply training the network on a specific class of adversarial examples might not form a sufficient defense strategy. Given this vulnerability of neural networks to deformations, we wish to study in future work how ADef can help for designing possible defense strategies. Furthermore, we also showed initial results on fooling adversarially trained networks. Remarkably, PGD trained networks on MNIST are more resistant to adversarial deformations than ADef trained networks. However, for this result to be more conclusive, similar tests on ImageNet will have to be conducted. We wish to study this in future work.

ACKNOWLEDGMENTS

The authors would like to thank Helmut Bölcskei and Thomas Wiatowski for fruitful discussions.

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

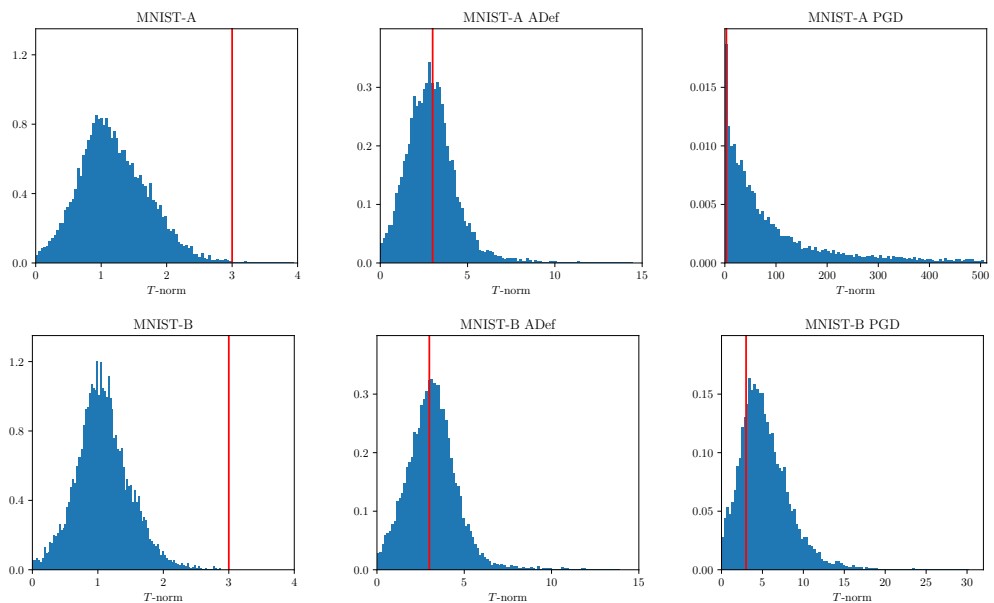

Figure 5: The (normalized) distribution of $\|\tau^*\|_T$ from the MNIST experiments. Deformations that fall to the left of the vertical line at $\varepsilon = 3$ are considered successful. The networks in the first column were trained using the original MNIST data, and the networks in the second and third columns were adversarially trained using ADef and PGD, respectively.

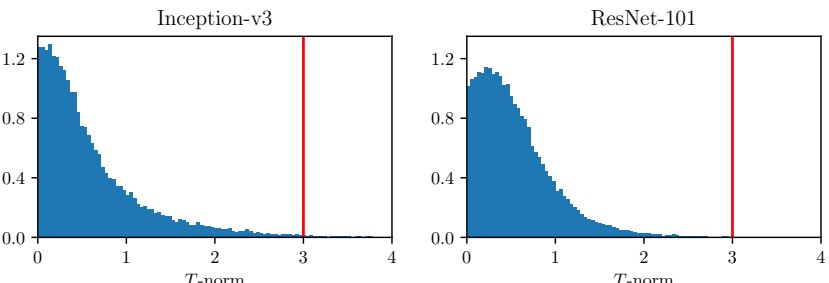

Figure 6: The (normalized) distribution of $\|\tau^*\|_T$ from the ImageNet experiments. Deformations that fall to the left of the vertical line at $\varepsilon = 3$ are considered successful.

## A  DISTRIBUTION OF VECTOR FIELD NORMS

Figures 5 and 6 show the distribution of the norms of the total deforming vector fields, $\tau^*$, from the experiments in section 3. For networks that have not been adversarially trained, most deformations fall well below the threshold of $\epsilon = 3$. Out of the adversarially trained networks, only MNIST-A trained against PGD is truly robust against ADef. Further, a comparison between the first column of figure 5 and figure 6 indicates that ImageNet is much more vulnerable to adversarial deformations than MNIST, also considering the much higher resolution of the images in ImageNet. Thus, it would be very interesting to study the performance of ADef with adversarially trained network for ImageNet, as mentioned in the Conclusion.

Table 3: The results of applying ADef to the images in the ILSVRC2012 validation set and the Inception model, using different values for the standard deviation $\sigma$ of the Gaussian filter. As before, we define ADef to be successful if $\|\tau^*\|_T \leq 3$.

| $\sigma$ | ADef success | Avg. $\|\tau^*\|_T$ | Avg. $\|r\|_\infty$ | Avg. # iterations |
|---|---|---|---|---|
| 0 | 99.12% | 0.5272 | 0.1628 | 5.247 |
| 1 | 98.94% | 0.5984 | 0.2039 | 4.050 |
| 2 | 95.91% | 0.7685 | 0.2573 | 3.963 |
| 4 | 86.66% | 0.9632 | 0.3128 | 4.379 |
| 8 | 67.54% | 1.1684 | 0.3687 | 5.476 |

## B  SMOOTH DEFORMATIONS

The standard deviation of the Gaussian filter used for smoothing in the update step of ADef has significant impact on the resulting vector field. To explore this aspect of the algorithm, we repeat the experiment from section 3 on the Inception-v3 model, using standard deviations $\sigma = 0, 1, 2, 4, 8$ (where $\sigma = 0$ stands for no smoothing). The results are shown in table 3, and the effect of varying $\sigma$ is illustrated in figures 7 and 8. We observe that as $\sigma$ increases, the adversarial distortion steadily increases both in terms of vector field norm and perturbation norm. Likewise, the success rate of ADef decreases with larger $\sigma$. However, from figure 8 we see that the constraint $\|\tau^*\|_T \leq 3$ on the total vector field may provide a rather conservative measure of the effectiveness of ADef in the case of smooth high dimensional vector fields.

## C  ADDITIONAL DEFORMED IMAGES

### C.1  MNIST

Figures 9 and 10 show adversarial deformations for the models MNIST-A and MNIST-B, respectively. The attacks are performed using the same configuration as in the experiments in section 3. Observe that in some cases, features resembling the target class have appeared in the deformed image. For example, the top part of the 4 in the fifth column of figure 10 has been curved slightly to more resemble a 9.

### C.2  IMAGENET

Figures 11 – 15 show additional deformed images resulting from attacking the Inception-v3 model using the same configuration as in the experiments in section 3. Similarly, figures 16 – 20 show deformed images resulting from attacking the ResNet-10 model. However, in order to increase variability in the output labels, we perform a targeted attack, targeting the label of 50th highest probability.

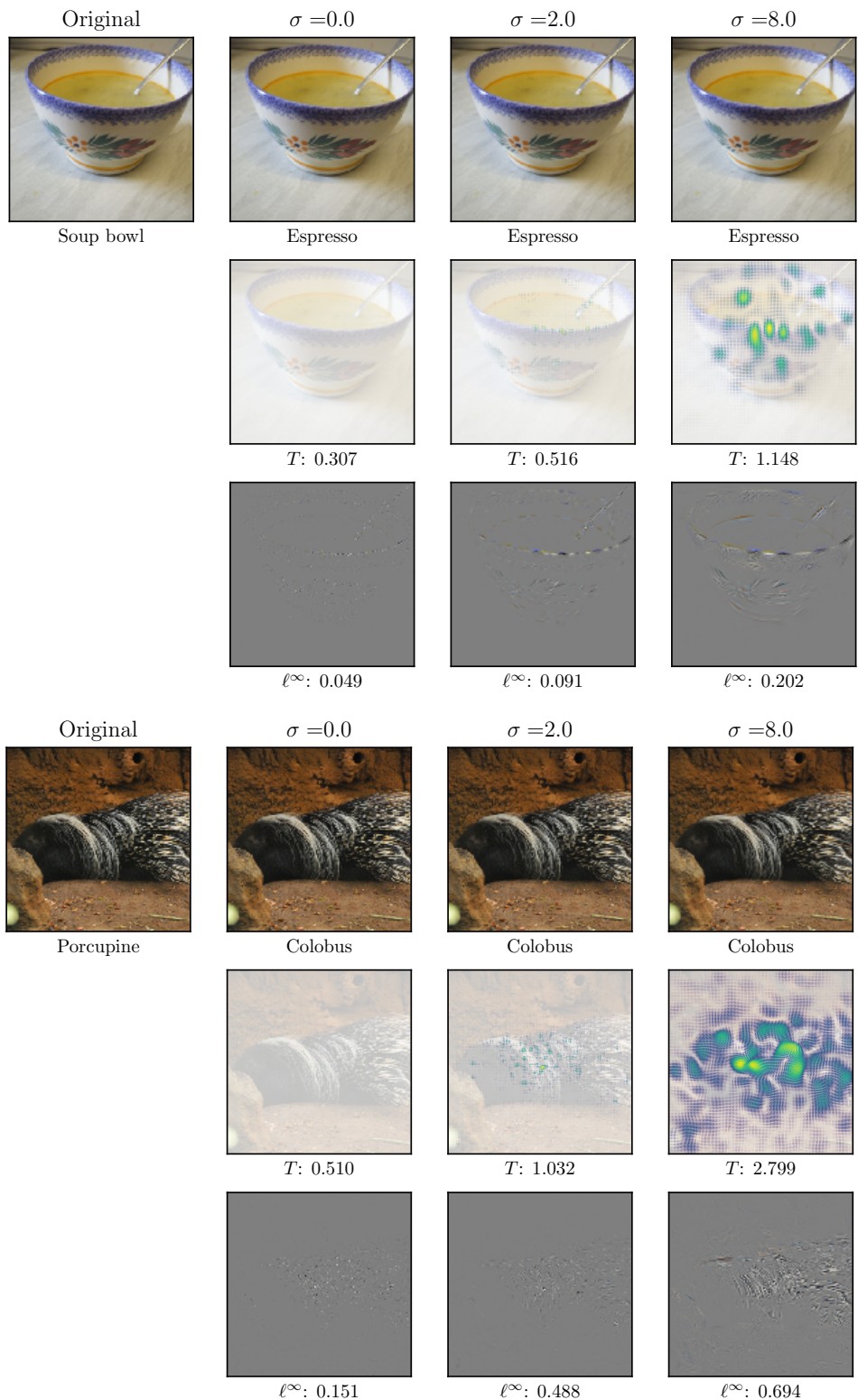

Figure 7: The effects of increasing the smoothness parameter $\sigma$ on adversarial deformations for Inception-v3. **First and fourth rows:** A correctly classified image and deformed versions. **Second and fifth rows:** The corresponding deforming vector fields and their $T$-norms. **Third and sixth rows:** The corresponding perturbations and their $\ell^\infty$ norms.

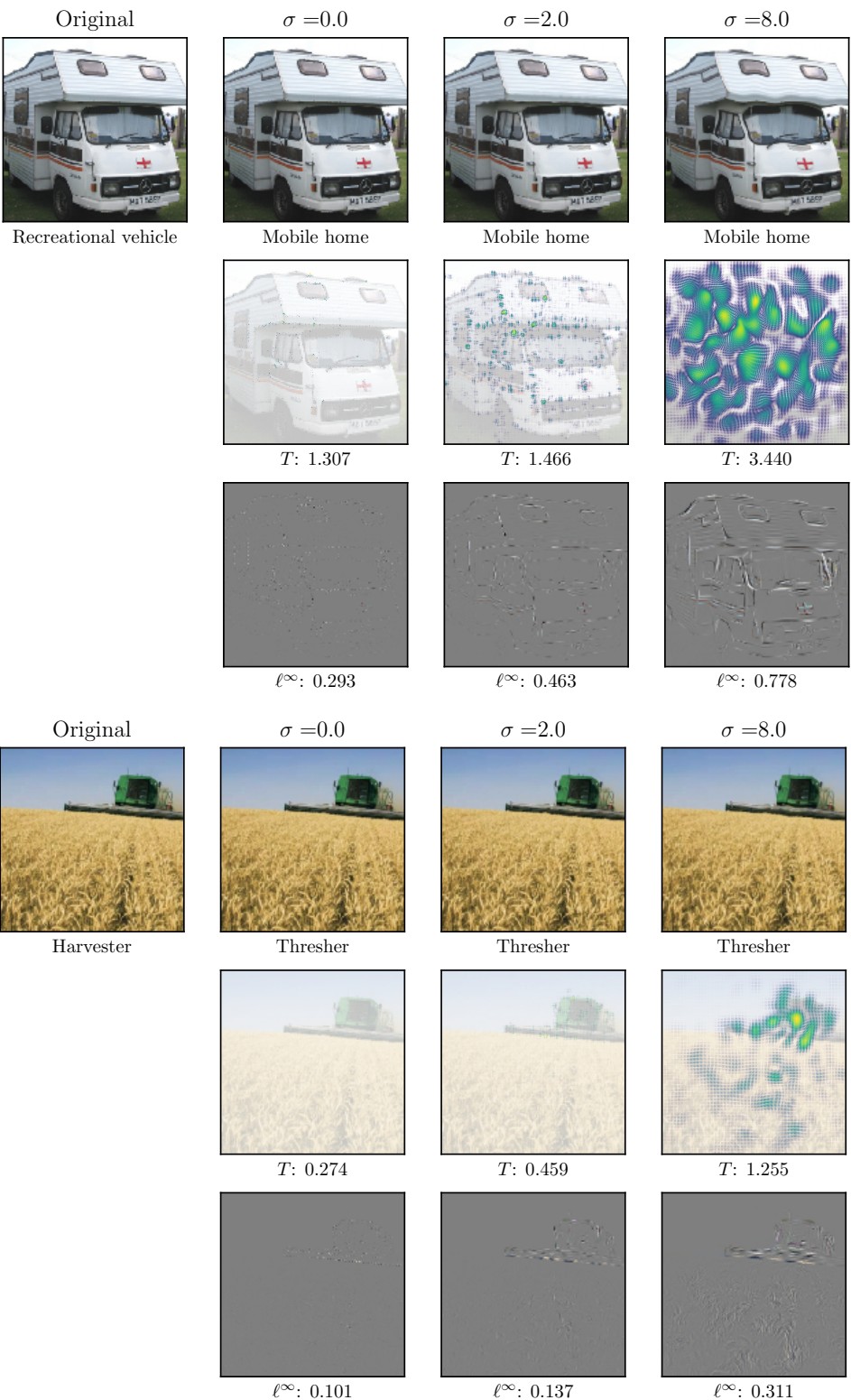

Figure 8: The effects of increasing the smoothness parameter $\sigma$ on adversarial deformations for Inception-v3. Note that according to the criterion $\|\tau^*\|_T \leq 3$, the value $\sigma = 8$ yields an unsuccessful deformation of the recreational vehicle.

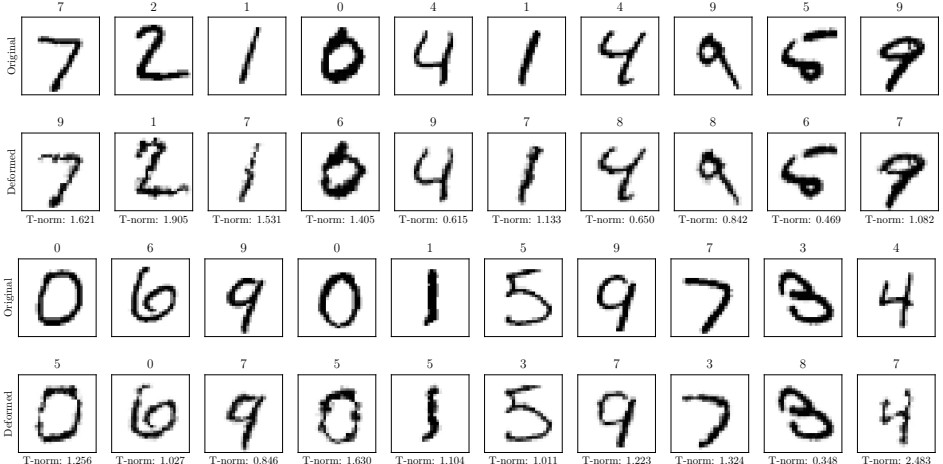

Figure 9: Adversarial deformations for MNIST-A. **First and third rows:** Original images from the MNIST test set. **Second and fourth rows:** The deformed images and the norms of the corresponding deforming vector fields.

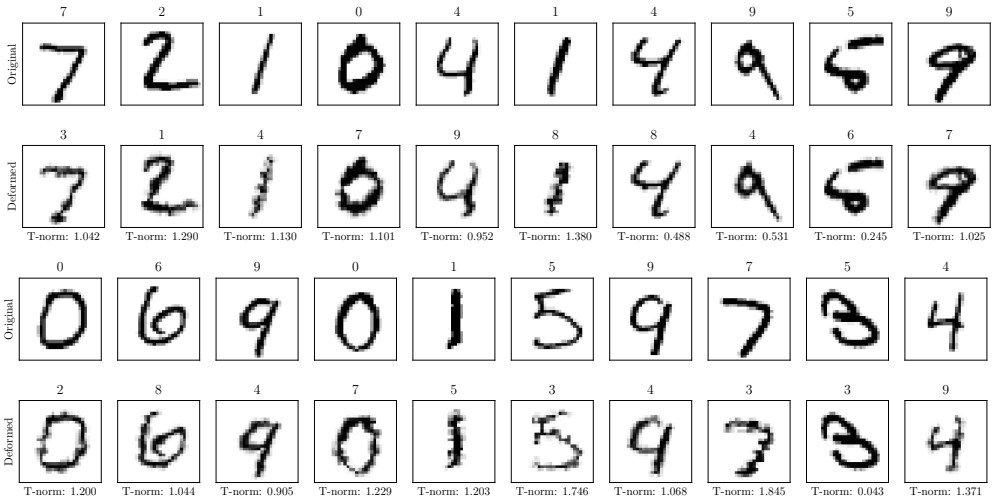

Figure 10: Adversarial deformations for MNIST-B. Note that image 9 in row 3 is misclassified, and is then deformed to its correct label.

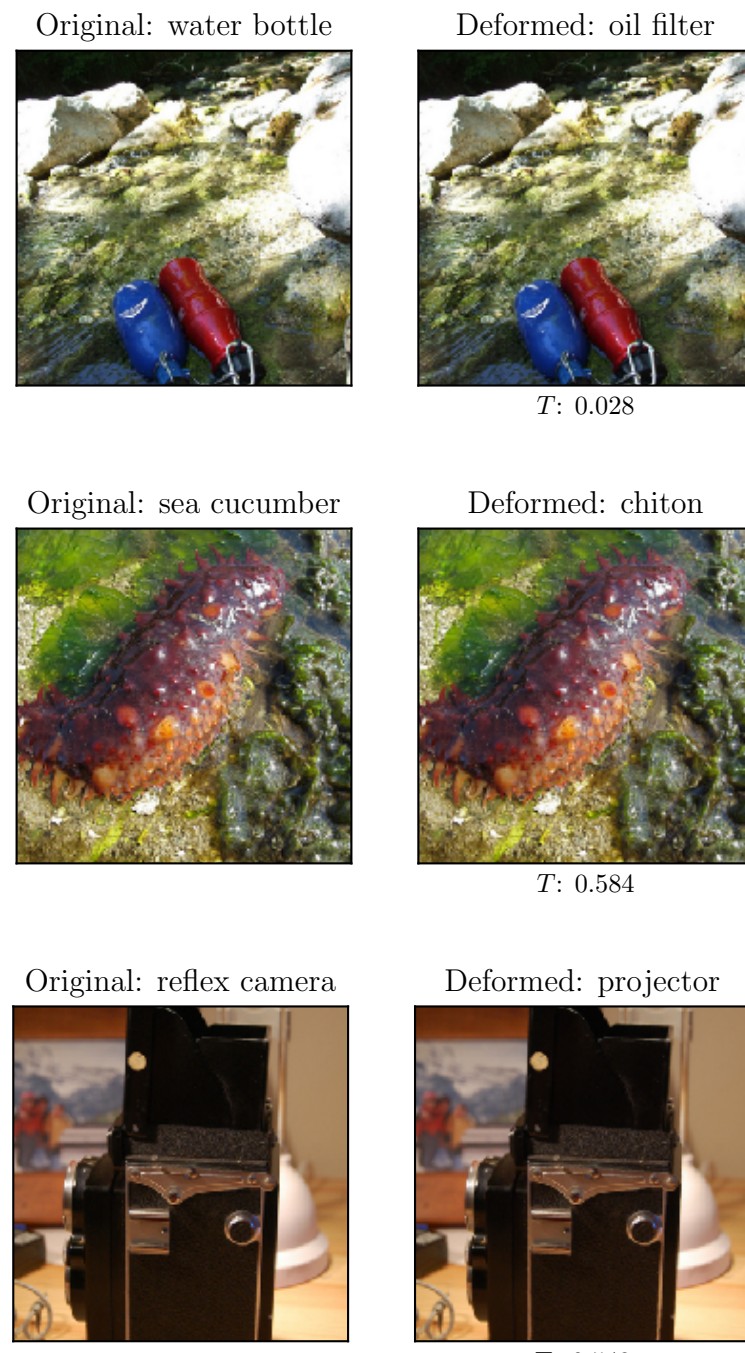

Figure 11: ADef attacks on the Inception-v3 model using the same configuration as in the experiments in section 3.

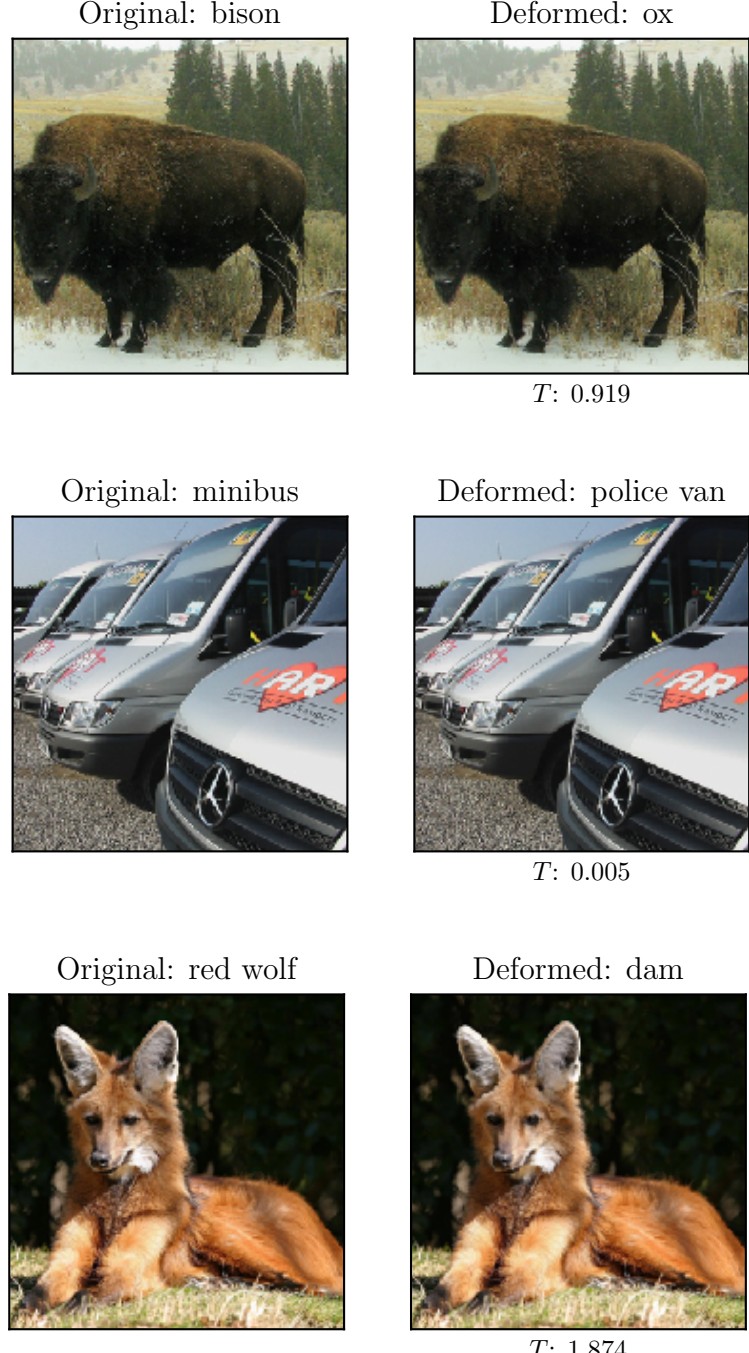

Figure 12: ADef attacks on the Inception-v3 model using the same configuration as in the experiments in section 3.

Original: water tower

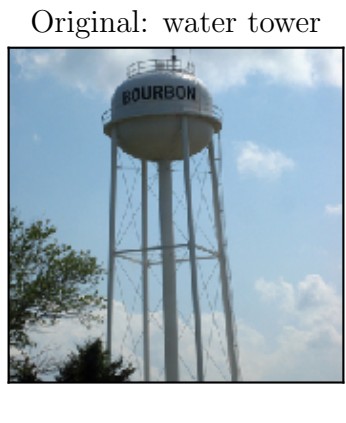

Deformed: chime

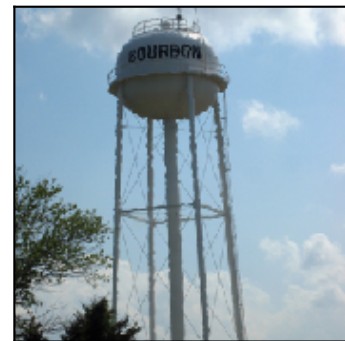

$T$: 2.791

Original: pelican

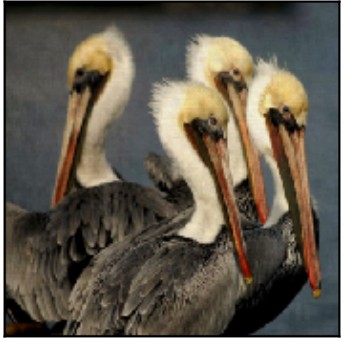

Deformed: bath towel

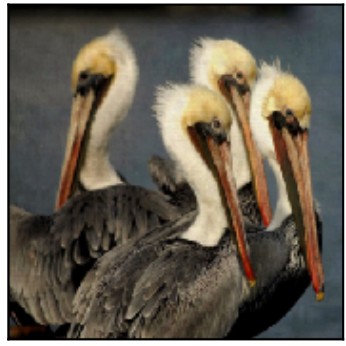

$T$: 0.906

Original: reflex camera

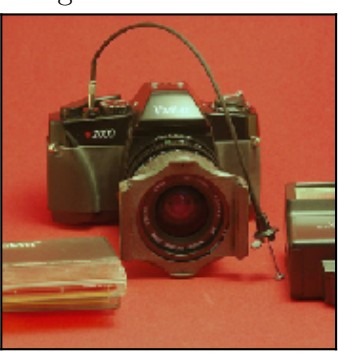

Deformed: Polaroid camera

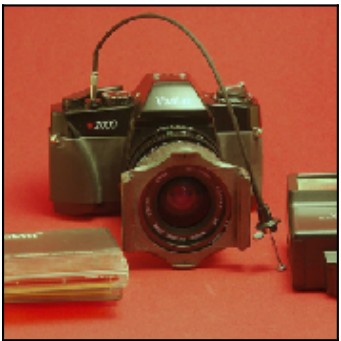

$T$: 0.061

Figure 13: ADef attacks on the Inception-v3 model using the same configuration as in the experiments in section 3.

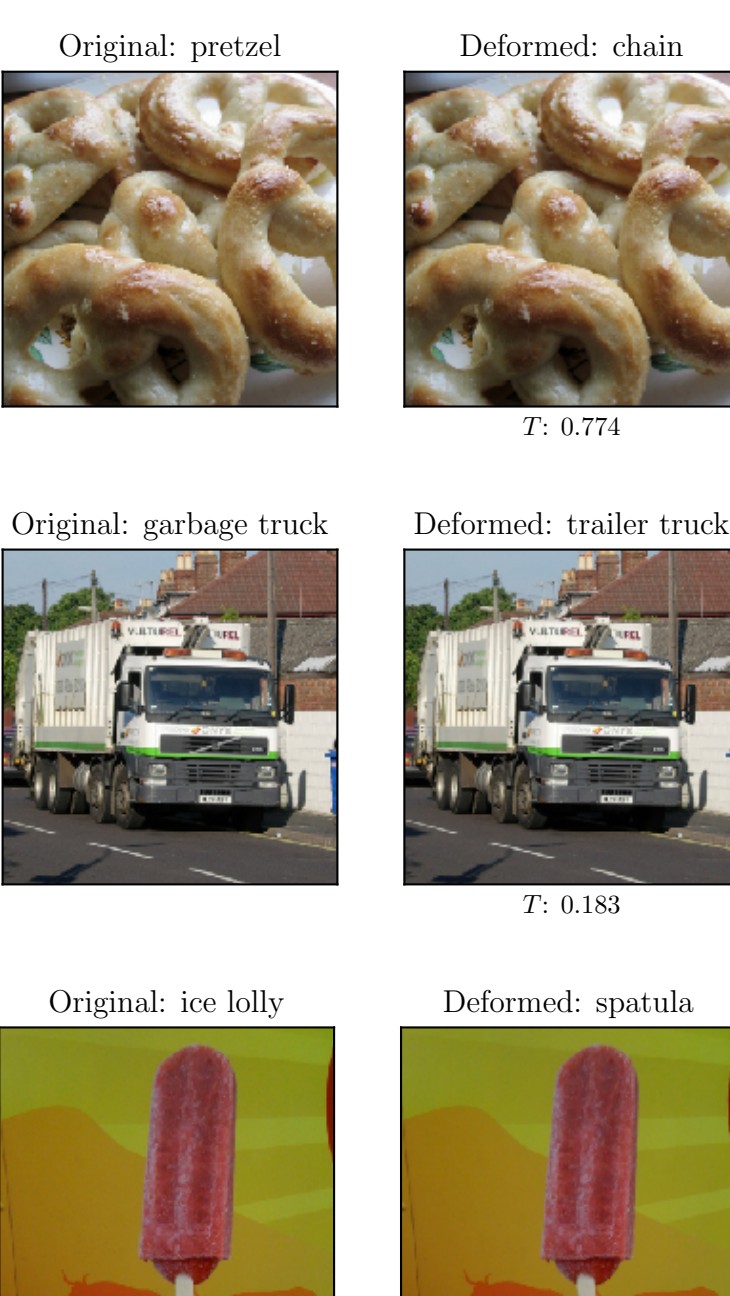

Figure 14: ADef attacks on the Inception-v3 model using the same configuration as in the experiments in section 3.

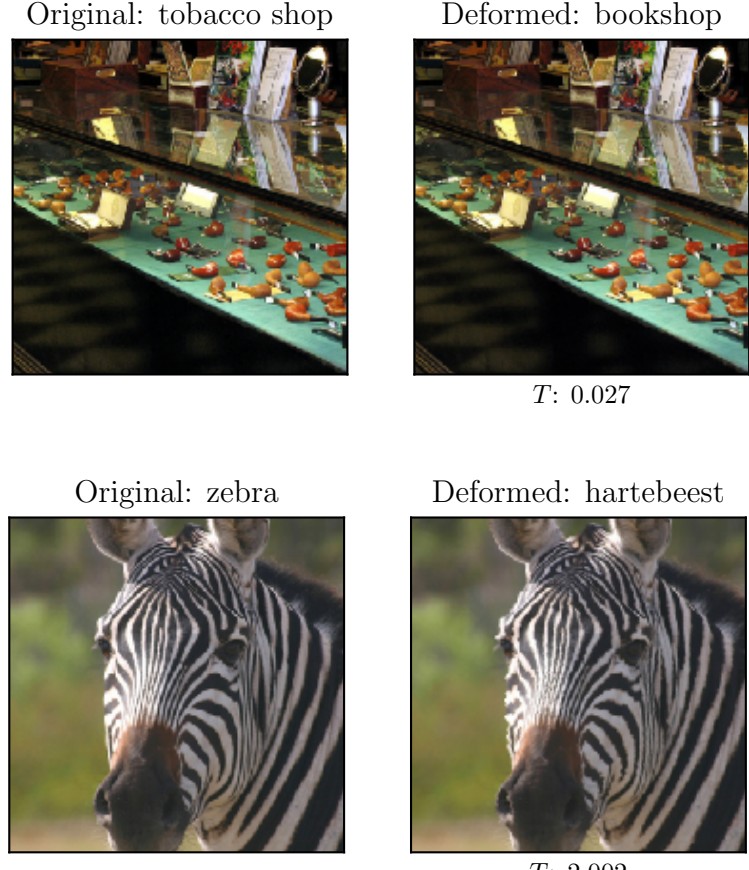

Figure 15: ADef attacks on the Inception-v3 model using the same configuration as in the experiments in section 3.

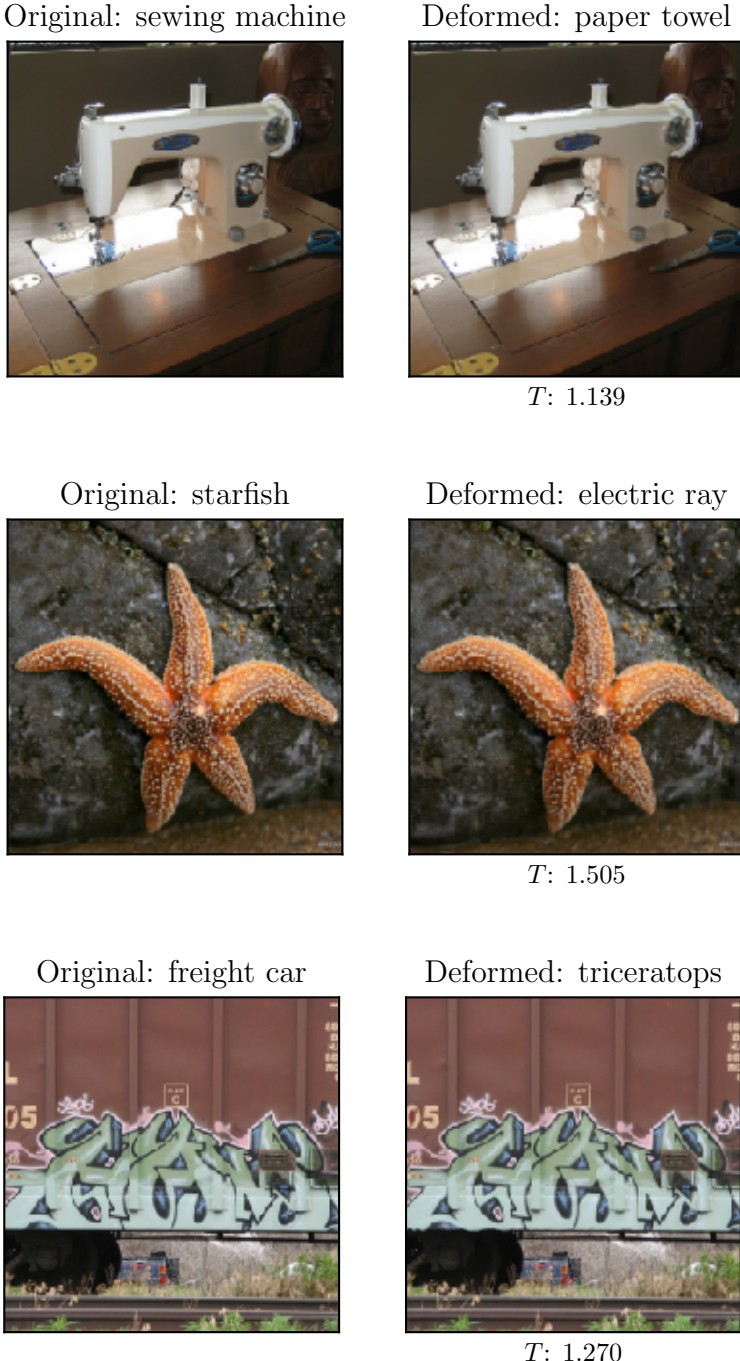

Figure 16: ADef attacks on the ResNet-101 model targeting the 50th most likely label.

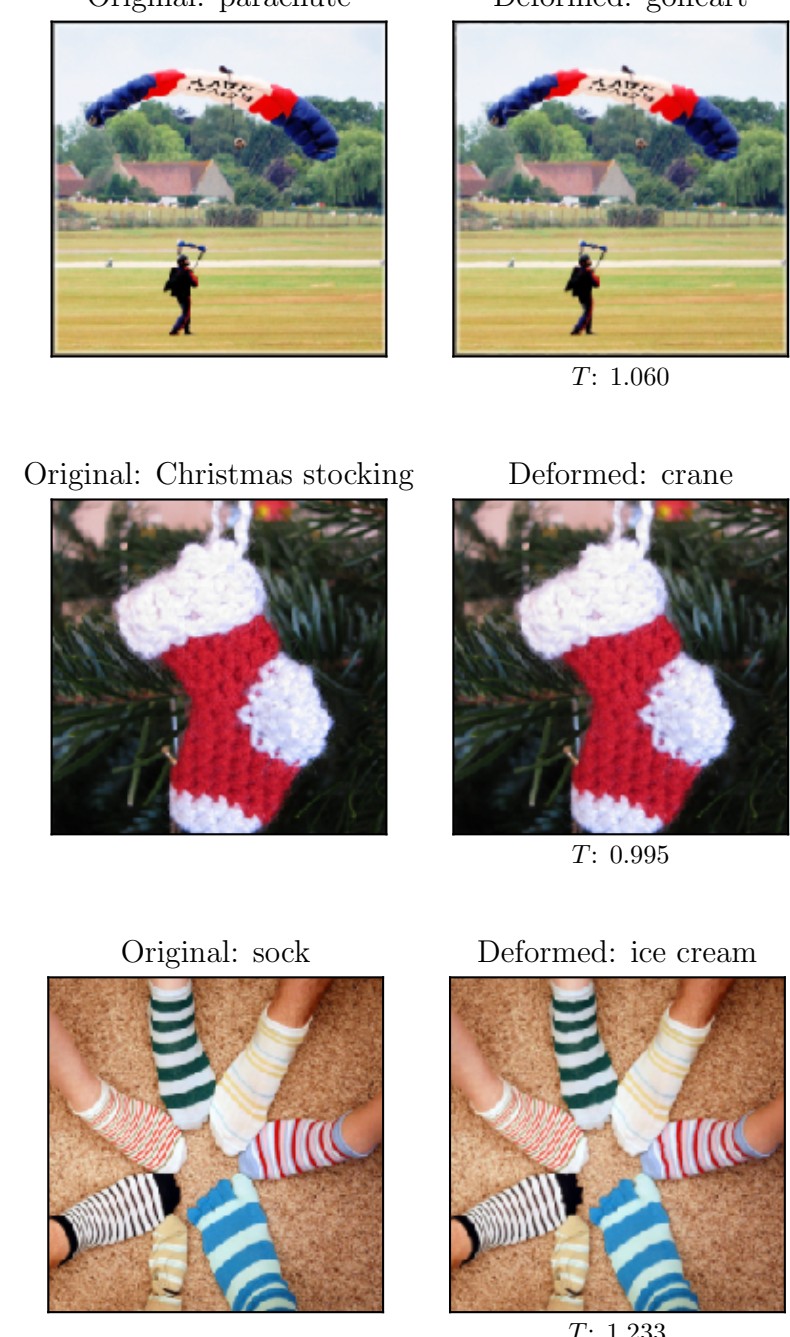

Figure 17: ADef attacks on the ResNet-101 model targeting the 50th most likely label.

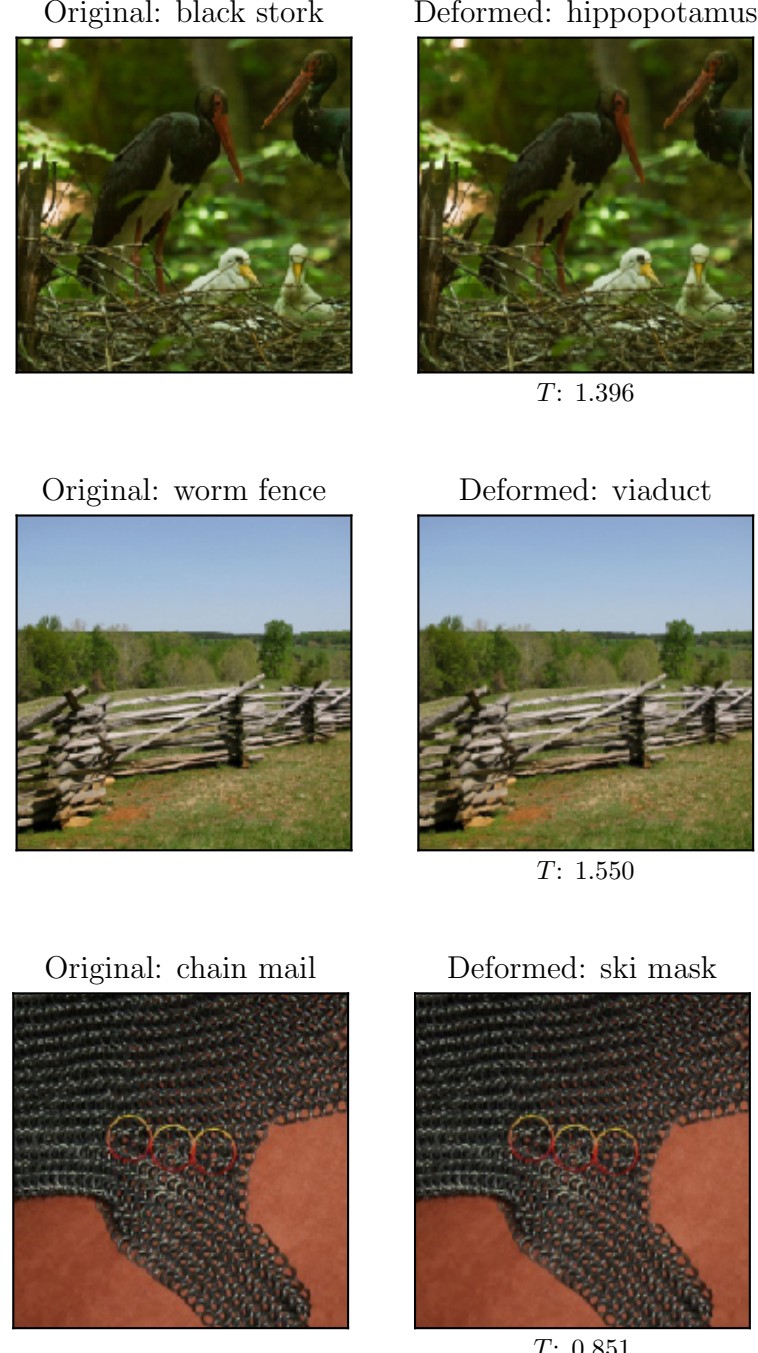

Figure 18: ADef attacks on the ResNet-101 model targeting the 50th most likely label.

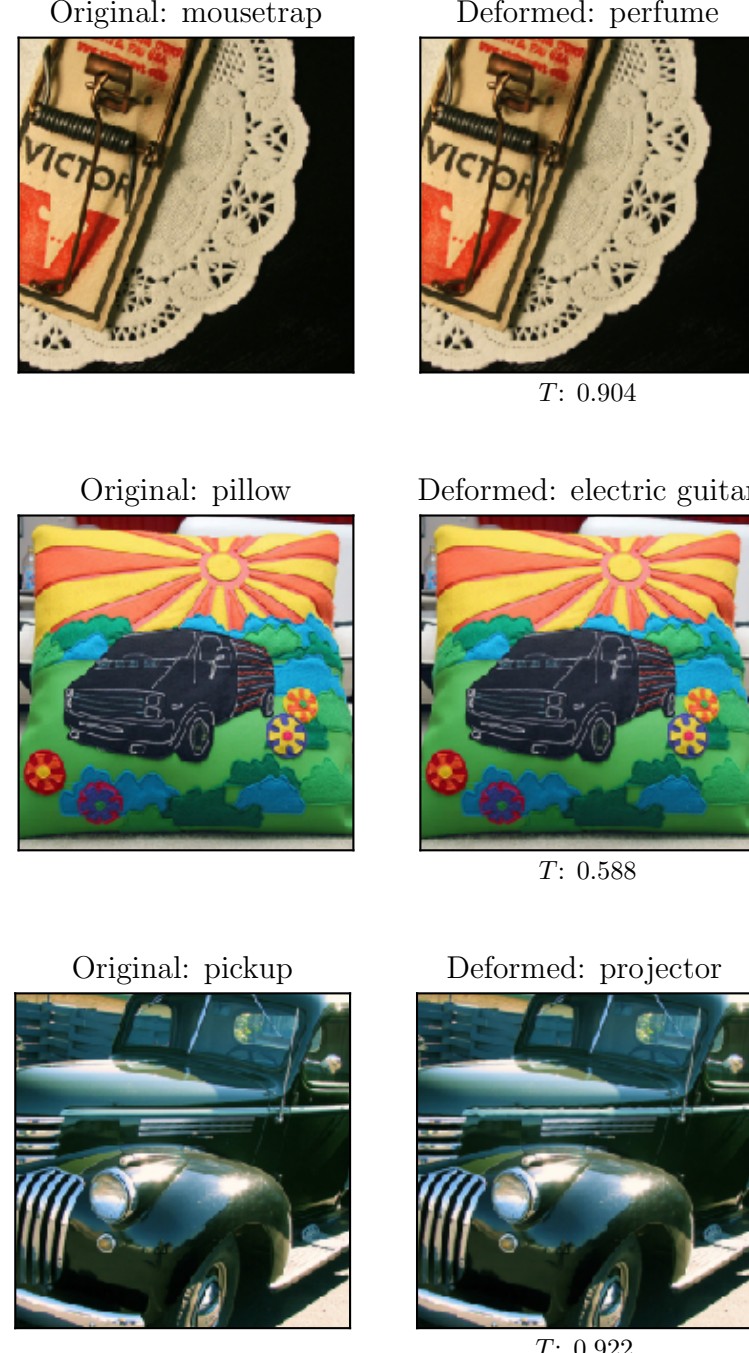

Figure 19: ADef attacks on the ResNet-101 model targeting the 50th most likely label.

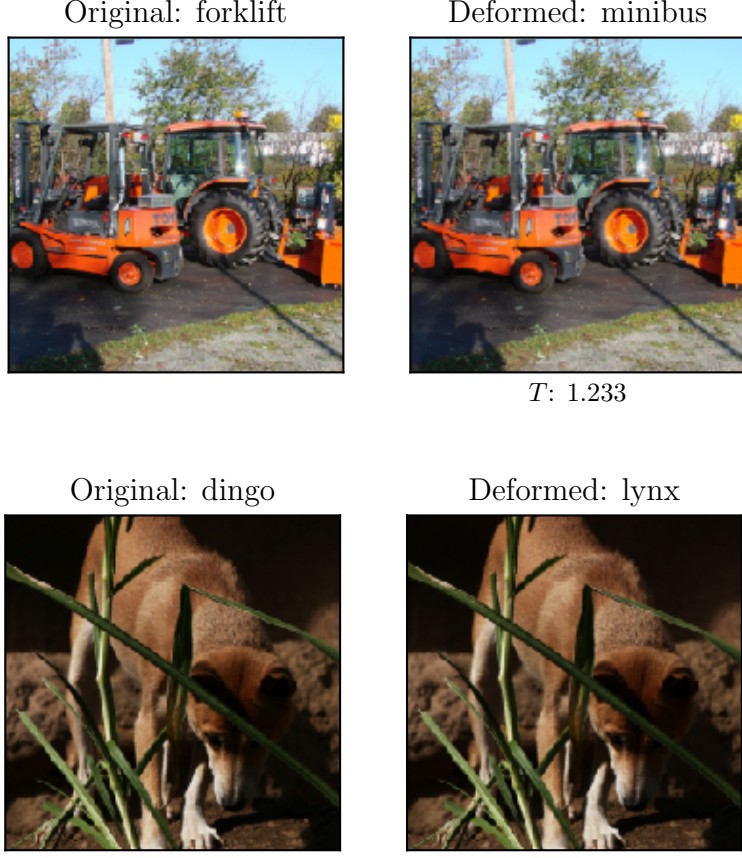

Figure 20: ADef attacks on the ResNet-101 model targeting the 50th most likely label.

