# OpenReview forum: "ADef: an Iterative Algorithm to Construct Adversarial Deformations"
_ICLR.cc/2019/Conference_

### Official Review · AnonReviewer2 · 2018-11-03
**The topic is interesting but the overall technical contribution looks weak.**

**Rating:** 6
**Confidence:** 3

**Review:**

The paper introduces an iterative method to generate deformed images for adversarial attack. The core idea is to perturb the correctly classified image by iteratively applying small deformations, which are estimated based on a first-order approximation step, until the image is misclassified. Experimental results on several benchmark datasets (MNIST, ImageNet) and commonly used deep nets (CNN, ResNet, Inception) are reported to show the power of adversarial deformations.

The idea of gradually adding deformations based on gradient information is somewhat interesting, and novel as far as the reviewer knows about. The method is clearly presented and the results are mostly easy to access.  However, the intuition behind the proposal does not make strong sense to the reviewer: since the main focus of this work is on model attack, why not directly (iteratively or not) adding random image deformations to fool the system? Particularly, the first-order approximation strategy (as shown in Eq.4 and Eq.5) is quite confusing. On one side (see Eq.4), the deformation \tau should be small enough in scale to make an accurate approximation. On the other side (see Eq. 5), \tau is required to be sufficiently large in order to generate misclassification. Such seemingly conflicting rules for estimating the deformation makes the proposed method less rigorous in math.
As another downside, the related adversarial training procedure is not fully addressed. The authors briefly discussed this point in the experiment section and provided a few numerical results in Table 2. These results, as acknowledged by the authors, do not well support the effectiveness of deformation adversarial attack and defense. In the meanwhile, the mentioned adversarial training framework follows straightforwardly from PGD (Madry et al. 2018), and thus the novelty of this contribution is also weak. More importantly, it is not clear at all, both in theory and algorithm, whether the advocated gradual deformation attack and defense can be unified inside a joint min-max/max-min learning formulation, as what PGD is rooted from.

Pros:

- The way of constructing deformation adversarial is interesting and novel
- The paper is mostly clearly organized and presented.

Cons:

- The motivation of approach is questionable.
- The related adversarial training problem remains largely unaddressed.
- Numerical study shows some promise in adversarial attack, but is not supportive to the related defense capability.

---

> ### Author Response · Authors · 2018-11-08
> **Response to Reviewer 2**
>
> Thank you, Reviewer 2. Your critique revolves around two issues, motivation of algorithm and adversarial training, both of which we believe can be addressed by the following clarification.
>
> Motivation of algorithm:
> 1. Deforming images by small random vector fields does not in general induce misclassification. This mirrors the fact that adding a random noise mask to an image usually does not change the predicted label (see for example [3]). For an adversarial attack to be effective, the image transformation has to be selected carefully.
> 2. Unless the derivative of the classifier in question is very small, Eq. 5 does not require the vector field \tau to be large. On the contrary, the experiments show that in the overwhelming majority of cases we find very small vector fields that satisfy this condition. In the derivation of the algorithm, we do not guarantee the existence of adversarial deformations. We simply say that if there exists a suitably small vector field that satisfies Eq. 5, then (by Eq. 4) the corresponding deformed image will be misclassified. In Eq. 7, we give a formula for a vector field that satisfies Eq. 5, and turns out to be small most of the time. This is a common line of reasoning in the literature of adversarial attacks: moving iteratively in the direction of the classifier’s gradient increases the chances of finding small adversarial transformations. We specifically point to the DeepFool attack for perturbations by Moosavi-Dezfooli et al. [2], which inspired our construction of deformations.
>
> Adversarial training:
> 1. When evaluating the effectiveness of a new adversarial attack, one may wonder if models that are specifically designed to resist adversarial examples are also robust to the new attack. To our knowledge, the PGD adversarial training procedure of Madry et al. [1] is the best defense against adversarial examples on MNIST. We show that this defense method is not sufficient to achieve robustness to adversarial deformations.
> 2. A natural question is whether this kind of adversarial training can be adapted to resist adversarial deformations. As a first step in that direction, we show that doing this in the most straightforward way, by replacing PGD with ADef in the training loop, does not yield better results. We do not propose this as a novel method for adversarial training. The current work focuses on introducing the new attack method, and it is beyond the scope of the paper to formulate rigorously a defense strategy to combat it.
>
> [1] A. Madry, A. Makelov, L. Schmidt, D. Tsipras, A. Vladu. Towards deep learning models resistant to adversarial attacks. ICLR2018.
> [2] S. M. Moosavi-Dezfooli, A. Fawzi, and P. Frossard. DeepFool: a simple and accurate method to fool deep neural networks. CVPR2016.
> [3] C. Szegedy, W. Zaremba, I. Sutskever, J. Bruna, D. Erhan, I. Goodfellow, R. Fergus. Intriguing properties of neural networks. arXiv:1312.6199.

---

> > ### Comment · AnonReviewer2 · 2018-11-16
> > **Discussion**
> >
> > Thank you for providing very detailed author response and paper revision. I found my main concerns reasonably clarified in this feedback, especially the concern on adversarial training. However, I am still a bit worried about solving Equation (5) in a way of unconstrained least-squares, which clearly does not guarantee to generate small deformation \tau as required for accurate first-order approximation. It is good to know from the response that in most cases in practice very small vector fields can be found to approximately satisfy Equation (5). My question (for discussion) here is: if this is the case, then why not imposing a box-constraint on \tau so as to explicitly enforce small deformation when solving (5)? I believe using such a box-constrained quadratic program could make the entire framework more rigorous in reasoning. Nevertheless, given that my major concerns are satisfactorily addressed and in view of the positive opinions from fellow reviewer, I will not be opposed to accepting the paper with further modifications. In the meanwhile, I also would like to hear from the author(s) about the above discussion point.

---

> > > ### Author Response · Authors · 2018-11-22
> > > **Further discussion**
> > >
> > > Thank you very much for the reconsideration. We are happy to engage in further discussion, and your question is interesting indeed. Our method focuses on finding an exact solution to Equation (5). This equation has many solutions, and in Equation (7) we choose the one that minimizes the l^2 norm of the vector field. One could replace the vector field update in (7) with a vector field satisfying (5) with minimal T-norm. We have been able to produce smaller vector fields in this way, however, in our experience the resulting vector fields tend to be visually less satisfying. This has to do with the vector directions being determined only by the signs of the entries from Equation (6).
> > >
> > > In this context, we would like to point out that it is not so clear when a deformation is too large, and thus it is difficult to decide on a box-constraint. We decided to reject vector fields of T-norm greater than 3 in order to have a credible, quantifiable measure of the success rate of ADef. In general, smooth deformations of size less than 3 should not change the semantic meaning of an MNIST image, but this condition can be relaxed immensely with higher dimension. Nevertheless, we chose to use the same criterion for ImageNet, simply because the vast majority of deformations fall below this very low threshold of 3. With this in mind, we think the entire distribution of the vector field norms are of interest, and hence we include Appendix A.

---

### Official Review · AnonReviewer3 · 2018-11-04
**interesting attack**

**Rating:** 7
**Confidence:** 3

**Review:**

In this paper, the authors proposed a new attack using deformation. The results are quite realistic to the naked eyes (at least for the example shown).  The idea is quite simple, generate small displacement and resample (interpolate) image until the label flips.

- I think this is a good contribution, It is a kind of attack we should consider.
- One thing which is good to consider is the type of interpolation. I believe the success rate would be different for linear versus say B-spline interpolation. Also, the width of the smoothing applied to the deformation field has an impact. The algorithm is straightforward, there is no reason to experiment with those.

- It is useful to report pixel displacement in Table 1. The reported values are not intuitive, the **average** displacement for Inception-v3 is 0.59.  Here is my back of envelope conversion of 0.59 which is probably off:

299 (# pixels of the smaller axis 299 for the Inception) x 1/2 (image are centered) x 0.59 =  88 pixels

This is huge! I think I am calculating something incorrectly because in Fig3,4 those displacements are not big.

- The results of Table 2 is interesting. Why a networked trained with PGD is more robust to ADef attack that a network trained adversarially with Adef?




Minor:
- The paper is a bit nationally convoluted for no good reason, the general idea is straightforward.

---

> ### Author Response · Authors · 2018-11-08
> **Response to Reviewer 3**
>
> We thank Reviewer 3 for the feedback. We would like to respond to the main points raised.
>
> With our configuration of smoothing and interpolation, we observed ADef to be very effective. We agree that quantitative experiments with different interpolation schemes would not have improved our contribution. However, experiments with different levels of smoothing may be of interest, and we will add them to the appendix soon (in addition to the example shown in Figure 2).
>
> To clarify the interpretation of Table 1, we have now included in the caption a reference to Equation 3, which defines the T-norm of vector fields. In the case of Inception, an average deforming vector field displaces all pixels by less than 0.59 pixels. That is only 0.59/299 = 0.2% of the image width.
>
> We agree that it is interesting that training with PGD provides better protection against ADef, than training with ADef. We find it quite remarkable that PGD training, which only considers small additive perturbations, can (to a degree) resist deformations, which in general correspond to large additive perturbations. However, we remark that it is unclear what the effect of ADef training is in terms of the geometry of the input space, while it is clear from Madry et al. [1] how PGD training has regularizing effects.
>
> [1] A. Madry, A. Makelov, L. Schmidt, D. Tsipras, A. Vladu. Towards deep learning models resistant to adversarial attacks. ICLR2018.

---

### Official Review · AnonReviewer1 · 2018-11-06
**Paper introducing an interesting new idea, that I would wish to see further developped**

**Rating:** 7
**Confidence:** 4

**Review:**

The paper proposes a new way to construct adversarial examples: do not change the intensity of the input image directly, but deform the image plane (i.e. compose the image with Id + tau where tau is a small amplitude vector field).

The paper originates from a document provably written in late 2017, which is before the deposit on arXiv of another article (by different authors, early 2018) which was later accepted to ICLR 2018 [Xiao and al.]. This remark is important in that it changes my rating of the paper (being more indulgent with papers proposing new ideas, as otherwise the novelty is rather low compared to [Xiao and al.]).

Pros:
- the paper is well written, very easy to read, well explained (and better formalized than [Xiao and al.]);
- the idea of deforming images is new (if we forget about [Xiao and al.]) and simple;
- experiments show what such a technique can achieve on MNIST and ImageNet. Interestingly, one can see on MNIST the parts of the numbers that the adversarial attack is trying to delete/create.

Cons:
- the paper is a bit weak, in that it is not very dense, and in that there is not much more content than the initial idea;
- for instance, more discussions about the results obtained could have been appreciated (such as my remark above about MNIST);
- for instance, a study of the impact of the regularization would have been interesting (how does the sigma of the Gaussian smoothing affect the type of adversarial attacks obtained and their performance -- is it possible to fool the network with [very] smooth deformations?);
- for instance, what about generating adversarial examples for which the network would be fully (wrongly) confident? (instead of just borderline unsure); etc.
- The interpolation scheme (how is defined the intensity I(x,y) for a non-integer location (x,y) within the image I) is rather important (linear interpolation, etc.) and should be at least mentioned in the main paper, and at best studied (it might impact the gradient descent path and the results);
- question: does the algorithm converge? could there be a proof of this? This is not obvious, as the objective potentially changes with time (selection of the current m best indices k of |F_k - F_l|). Also, the final overshoot factor (1+eta) is not very elegant, and not guaranteed to perform well if tau* starts being not small compared to the second derivative (i.e. g''.tau^2 not small) while I guess that for image intensities, spatial derivatives can be very high if no intensity smoothing scheme is used.
- note: the approximation tau* = sum_i tau_i (section 2.3) does not stand in the case of non-small deformations.
- still in section 2.3, I do not understand the statement "given that \nabla f is moderate": where does this property come from? or is "given" meant to be understood as "provided..." (i.e. under the assumption that...)?
- computational times could have been given (though I guess they are reasonable).

Other remarks:
- suggestion: I find the "slight abuse of notation" (of confusing the derivative with the gradient) a bit annoying and suggest to use a different symbol, such as \nabla g. This could be useful in particular in the following perspective:
- Mathematical side note: the "gradient" of a functional is not a uniquely-defined object in that it depends on the metric chosen in the tangent space. More clearly: the space of small deformations tau comes with an inner product (here L2, but one could choose another one), and the gradient \nabla g obtained depends on this inner product choice M, even though the derivative Dg is the same (they are related by Dg(tau) = < \nabla_M g | tau >_M for any tau). The choice of the metric can then be seen as a prior over desired gradient descent paths. In the paper, the deformation fields get smoothed by a Gaussian filter at some point (eq. 7), in order to be smoother: this can be interpreted as a prior (gradient descent paths should be made of smooth deformations) and as an associated inner product change (there do exist a metric M such that the gradient for that metric is \nabla_M g = S \nabla_L2 g). It is possible to favor other kind of deformations (not just smooth ones, but for instance rigid ones, etc. [and by the way this could make the link with "Manitest: Are classifiers really invariant?" by Fawzi and Frossard, BMVC 2015, who observe that a rigid motion can affect the classifier output]). If interested, you can check "Generalized Gradients: Priors on Minimization Flows" by Charpiat et al. for general inner products on deformations (in particular favoring rigid motion), and "Sobolev active contours" by Sundaramoorthi et al. for inner products more dedicated to smoothing (such as with the H1 norm).
- Note: about the remark in section 3.2: deformation-induced transformations are a subset of all possible transformations of the image (which are all representable with intensity changes), so it is expected that a training against attacks on the intensity performs better than a training against attacks on spatial deformations.

---

> ### Author Response · Authors · 2018-11-15
> **Response to Reviewer 1 (1/2)**
>
> Dear Reviewer 1, thank you for the detailed feedback. Its positive nature is much appreciated. In the following, we hope we can address each of your concern.
>
> We believe that the idea of moving from perturbations to deformations is substantial. In addition, the derivation of ADef is much less straightforward than the derivation of algorithms for adversarial perturbations because of the nonlinearity of the problem, as is evident from the mathematical details included in Appendix C. To address your specific comments on the discussion in the paper:
>
> 1. Your comment “one can see on MNIST the parts of the numbers that the adversarial attack is trying to delete/create” is an interesting observation. We would be happy to point this out in the text. It would most appropriately fit in Appendix B.1, with the images that show this effect. We will add to appendix B.1: “Observe that in some cases, features resembling the target class have appeared in the deformed image. For example, the top part of the 4 in the fifth column of figure 8 has been curved slightly to more resemble a 9.”
>
> 2. Showing quantitative results for varying width of the Gaussian filter is certainly of interest. We are preparing a short section on this issue, to be included in the appendices. These changes will be implemented as soon as possible, and the anonymized document updated.
>
> 3. One could come up with a variation of ADef to induce high confidence adversarial deformations. However, we consider targeted ADef to be the more important variant of our algorithm. In order not to complicate the presentation, we choose to discuss only the latter.
>
> 4. It should be stated explicitly in the paper that we use bilinear interpolation in our implementation. We do not expect the choice of interpolation to be of much consequence for high dimensional images, and including a study of this choice runs the risk of obscuring the main purpose of the paper. We will change the MNIST and ImageNet paragraphs on page 5 as follows: “It performs smoothing by a Gaussian filter of standard deviation 1/2, uses bilinear interpolation to obtain intermediate pixel intensities, and it overshoots [...]” and “It employs a Gaussian filter of standard deviation 1, bilinear interpolation, and an overshoot [...]”.
>
> 5. The question of convergence is interesting. However, we believe that the clear mathematical motivation and the effectiveness of ADef against existing models justifies the formulation of the algorithm sufficiently. Given the changes in the objective function you have noted, and potential nonlinearity in the update step we do not expect proof of convergence to be straightforward, and such an investigation might unnecessarily complicate the simple message of the paper: there exist small adversarial deformations, and an efficient way to find them.
>
> 6. We agree that the overshoot factor (1+\eta) is not very elegant, and that is why we only invoke it when ADef converges to a decision boundary. In practice, it works well to avoid ambiguous predictions. Likewise, in practice \tau* = \sum_i \tau_i is a helpful proxy to quickly estimate the total vector field, and relies on the (observed) facts that the individual steps are small and that few iterations are needed.
>
> 7. Thank you for pointing out the phrasing of the remark at the end of section 2.3. It should be understood as "provided that \nabla f is moderate", and will be fixed in the revision.
>
> 8. We do not have timing of the experiments, but it is indeed reasonable, and large scale experiments on high dimensional images are not prohibitive. We would like to point out that our code is available online, albeit not on an anonymous platform.

---

> > ### Author Response · Authors · 2018-11-15
> > **Response to Reviewer 1 (2/2)**
> >
> > 9. We are fully aware that this abuse of notation may create confusion. However, we reckon that most readers will prefer this formulation than a more correct, but more involved, derivation. As mentioned in the introduction, the interested reader may find all mathematical details in [1], now in Appendix C (to maintain anonymity). In particular, section C.1.2 contains the rigorous derivation, where the derivative and the gradient are kept distinct.
> >
> > 10. Many thanks for your comments on the link between the choice of the metric and the prior implicitly put in the deformations. We feel that this aspect requires a thorough analysis, which would go much beyond the scope of this paper, but would certainly be a very interesting topic for future research.
> >
> > 11. Thank you for the reference to Fawzi and Frossard [2], which should be cited as relevant work in the introduction. We will add the following at the very end of the introduction: “[...] performance of existing DNNs, and Fawzi & Frossard (2015), in which the authors introduce a method to measure the invariance of classifiers to geometric transformations.”
> >
> > 12. We do not think it is clear that a model trained against intensity perturbations is expected to perform better against deformations. It is true that a deformed image, y = x^\tau, can be written as a sum of the original image and a perturbation, y = x + r. However, the perturbation r is large in general (using the l^infinity norm), while adversarial training only takes into account small perturbations.
> >
> > [1] Author Anonymous. Adversarial perturbations and deformations for convolutional neural networks, 2017.
> > [2] A. Fawzi and P. Frossard. Manitest: Are classifiers really invariant? BMVC2015.

---

> > > ### Comment · AnonReviewer1 · 2018-11-20
> > > **Discussion**
> > >
> > > Thank you for your detailed answer to all of my remarks.
> > > I appreciate in particular the supplementary experiments with more regular deformations (higher sigma), though the results are difficult to interpret.
> > > I am still a bit disappointed by answer 4 (in that I guess it is pretty straightforward to adapt the algorithm to make it easier to analyze), and by the lack of additional discussions or interpretations of results in general in the main paper.
> > > I somehow agree with answer 12.
> > > Overall I will keep the same rating.

---

### Author Response · Authors · 2018-11-15
**Paper revision**

We thank all reviewers for their helpful input. We have now updated our submission accordingly. The revised version includes a short appendix which explores the effect of using a wider Gaussian filter for smoothing the deforming vector fields. Other changes are minor clarifications that are detailed in our responses to the reviewers.

---

### Meta-Review · Area_Chair1 · 2018-12-16
**Area chair recommendation**

**Confidence:** 4
**Recommendation:** Accept (Poster)

**Metareview:**

The submission proposes a method to construct adversarial attacks based on deforming an input image rather than adding small peturbations.  Although deformations can also be characterized by the difference of the original and deformed image, it is qualitatively and quantitatively different as a small deformation can result in a large difference.

On the positive side, this paper proposes an interesting form of adversarial attack, whose success can give additional insights on the forms of existing adversarial attacks.  The experiments on MNIST and ImageNet are reasonably comprehensive and allow interesting interpretation of how the image deforms to allow the attack.  The paper is also praised for its clarity, and cleaner formulation compared to Xiao et al. (see below).  Additional experiments during rebuttal phase partially answered reviewer concerns, and provided more information e.g. about the effect of the smoothness of the deformation.

There were some concerns that the paper primarly presents one idea, and perhaps missed an opportunity for deeper analysis (R1).  R2 would have appreciated more analysis on how to defend against the attack.

A controversial point is the relation /  novelty with respect to Xiao et al., ICLR 2018.  As e.g. pointed out by R1: "The paper originates from a document provably written in late 2017, which is before the deposit on arXiv of another article (by different authors, early 2018) which was later accepted to ICLR 2018 [Xiao and al.]. This remark is important in that it changes my rating of the paper (being more indulgent with papers proposing new ideas, as otherwise the novelty is rather low compared to [Xiao and al.])."

On the balance, all three reviewers recommended acceptance of the paper.  Regarding novelty over Xiao et al., even ignoring the arguable precedence of the current submission, the formulation is cleaner and will likely advance the analysis of adversarial attacks.